**Technical Note:**

**Open-paleo-data implementation pilot – The PAGES 2k special issue**

Darrell S. Kaufman[1] and PAGES 2k special-issue editorial team*

[1] School of Earth Sciences & Environmental Sustainability, Northern Arizona University, Flagstaff, USA.

* A full list of authors and their affiliations appears at the end of the paper.

*Correspondence to*: D. Kaufman (darrell.kaufman@nau.edu)

**Abstract.** Data stewardship is an essential element of the publication process. Knowing how to enact data polices that are described only in general terms can be difficult, however. Examples are needed to model the implementation of open-data polices in actual studies. Here we explain the procedure used to attain a high and consistent level of data stewardship across a special issue of the journal, *Climate of the Past*. We discuss the challenges related to (1) determining which data are essential for public archival, (2) using data generated by others, and (3) understanding data citations. We anticipate that open-data sharing in paleo sciences will accelerate as the advantages become more evident and as practices that reduce data loss become the accepted convention.

## 1 Introduction

The benefit of open data for accelerating scientific discovery and safeguarding scientific integrity is widely recognized (e.g., Michener, 2015), but the procedures for making data readily available for future use are still being developed. The paleo-science community has a long history of pooling data to advance the understanding of global change. On the other hand, a disparagingly low proportion of paleo sciences articles published today include ready access to the underlying data and essential metadata. In a recent survey, only 25% of earth scientists report that they transfer their data to public repositories (Stuart et al., 2018). The paleo-science community is currently in the early stages of adopting the relatively new open-data policies established by major publishers[1,2], research funders[3,4], and government agencies[5] (citations selected from among many). Few examples are available that model best practices for open paleo data as part of scientific publication, and best practices are evolving rapidly along with the revolution in cyber-enabled data sharing. Considering the vast variety of data types inherent in paleo sciences, knowing how to apply generalized data policies in real situations is not always obvious. Funding agencies and journals have set guidelines and requirements, but scientific communities must develop the protocols

---

[1] https://publications.agu.org/author-resource-center/publication-policies/data-policy/
[2] http://www.nature.com/authors/policies/availability.html
[3] https://www.nsf.gov/bfa/dias/policy/dmp.jsp
[4] https://erc.europa.eu/funding-and-grants/managing-project/open-access
[5] https://www2.usgs.gov/datamanagement/share.php

to make data readily and intelligently reusable (Kattge et al., 2014; McNutt et al., 2016). Motivation from the research community is needed to drive the open-data revolution (Molloy, 2011; Lemprière, 2017).

Transferring data to a public repository as soon as they are generated is ideal in many situations as part of a
project's comprehensive data management[6]; nevertheless, publication will always be a key high-value stage for data stewardship. This is because publications are …

… *incentivised*. Compliance with policies can be enforced by editors who determine when papers are ready for publication. There are few other points of control and reward associated with data stewardship.

… *the last stop*. Publication typically marks the termination of a study and is often the final pragmatic point in the life cycle
of a study to transfer the underlying data to a repository before the familiarity with the data and the incentive to archive them fade. Without proper curation in a public repository, the ability to locate data diminishes rapidly after publication, even for articles that had purported to make the associated data available (Vines et al., 2014).

… *quality controlled*. As authors analyse data in final preparation of their publication, errors might be exposed. Following submission, peer reviewers and editors of a journal article can help to double check the accuracy of the data and metadata as
part of the review and production process.

… *designed to maximize exposure*. Authors striving to enhance the impact and visibility of their study are receptive to input from peer reviewers and journal editors who can help guide authors toward making their data more easily discoverable and reusable. Articles that make the underlying data easily discoverable and reusable have been shown to benefit from a higher rate of citation (Piwowar and Vision, 2013).

… *required before a peer-reviewed interpretation (expert knowledge) is attached to a dataset*. For many studies, especially in paleoclimatology, the value of the underlying data is closely related to their interpretation. The most important data and metadata are typically those that are associated with a publication that describes them. Peer review can aid the interpretation of data and can help authors to identify the essential metadata. Encoding such peer-reviewed expert knowledge into an archived dataset is not possible prior to publication, but is necessary to facilitate the intelligent reuse of the data.

… *a means for data rescue*. Most paleoclimate studies compare their results with those from related study areas or data types. Often in paleo sciences, the digital data from the previously published studies are not available through public repositories. If the comparison with previous studies is the basis for a major conclusion, such as for a synthesis study, the authors of the succeeding publication can serve as data stewards by facilitating the transfer of data from previous publications to a public repository, with credit given to the original data generator. As part of this data rescue, authors can
attach relevant metadata to valued previously published datasets to enable their discoverability and intelligent reuse.

---

[6] https://www.dataone.org/sites/all/documents/DataONE_BP_Primer_020212.pdf

As the paleo community's data resources continue to grow, compilations that summarize and curate large datasets are becoming increasingly useful (Kaufman, 2017). The Past Global Changes (PAGES) project is an international, community-driven effort that facilitates the development of data-intensive synthesis products in paleo sciences, and promotes long-term care of the community's data resources. PAGES research is primarily conducted by working groups whose primary scientific goals are often based on an original global-scale data synthesis product (e.g., PAGES 2k Consortium, 2017a,b). Themes that involve multiple working groups are coordinated through PAGES' "integrative activities," of which data stewardship[7] is currently one. It has developed guidelines for open-data sharing[8] and has facilitated the creation of community-based data products that comprise well-described global datasets accessible through public repositories.

The purpose of this technical note is to describe the procedure used to implement the PAGES' data-sharing policy and to attain a high and consistent level of data stewardship throughout the special issue of *Climate of the Past*, "Climate of the past 2000 years: regional and trans-regional syntheses[9]," hereafter, the PAGES 2k special issue. This same procedure was subsequently implemented through the PAGES Young Scientists Meeting[10] special issue of the journal. We highlight key challenges and approaches to overcoming the challenges. Rather than discussing the merits of open-data sharing, we focus on the implementation of open-data policies. Readers may refer to the interactive discussion[11] associated with this note for comments and replies that include concerns about, and support for, sharing data. This technical note contributes to a large number of related efforts now underway and expanding across the earth sciences to promote open-data principles. Prominent among these is the "Enabling FAIR Data Project[12]" a major international effort moving forward earnestly (Stall et al., 2018) to promote data that are Findable, Accessible, Interoperable and Reusable (FAIR; Wilkinson et al., 2016), with many journals preparing to switch from open-data policies that are recommended to those that are required. This note documents how, as editors and reviewers, we enacted such a switch and what we learned in the process (Fig. 1).

## 2 Procedure for implementing the open-paleo-data policy

### 2.1 Reaffirm the intent and goals of relevant policies

The editorial team of the PAGES 2k special issue viewed the compendium as an opportunity to work with motivated authors to model best practices for open data associated with publication, despite the additional work required for both authors and editors, and despite the lack of an established roadmap. To begin, the goals of the data stewardship activity were discussed with the journal publisher, Copernicus Publications, who endorsed our enforcement of the journal's data policy. A notice

---

[7] http://pastglobalchanges.org/ini/int-act/data-stewardship
[8] http://pastglobalchanges.org/my-pages/data/data-guidelines
[9] https://www.clim-past.net/special_issue841.html
[10] https://www.clim-past.net/special_issue912.html
[11] https://www.clim-past-discuss.net/cp-2017-157/
[12] http://www.copdess.org/home/enabling-fair-data-project/

was sent to prospective authors of the special issue to explain the rationale and process for the data stewardship effort and to remind them of the relatively high standard as stated in the journal's[13] and PAGES'[8] data policies. Most importantly, rather

than *recommending* that the underlying data be made available, papers accepted to the special issue were *required* to make the data available. This critical point could not be reiterated enough, even though the PAGES 2k community has long recognized and reaffirmed the requirement to make all data used in its products available publicly. As journals move toward enforcing open-data practices, the specific requirements will need be made explicit as part of the information to prospective authors.

**2.2 Review manuscripts for data accessibility**

Submissions that did not include the required data availability statement were returned to authors with guidelines on how to prepare the statement prior to resubmitting. Once accepted to the Discussion phase, the editorial team reviewed the data contained in each manuscript. Generally, the most important data were presented in figures and tables, providing a focus for the data review. The editorial team identified the most relevant datasets of two general types: (1) data described in previous

publications, including data-reanalysis products, and third-party unpublished data ("input data"), and (2) data generated as an outcome of the study ("output data"). The editorial team used the information presented in the data availability statement and elsewhere in the manuscript to locate the datasets as described, sometimes unsuccessfully. Missing or inaccessible data were specified in a data-review comment, which was prepared by the editorial team, then posted as part of the public interactive discussion. Authors were asked to transfer all datasets that were not easily accessible to a community-recognized public data

repository, and to cite the dataset using the persistent identifier issued by the repository. Authors were asked to reply to the data-review comment as part of the public interactive discussion, as a matter of record for the publication. The data-review comments and author replies for each Discussion paper of the PAGES 2k special issue are available at the journal's website[7]; readers interested in the types and level of guidance provided by us to the authors are encouraged to read our data-review comments, or those from the PAGES Young Scientists Meeting[10] special issue of the journal. Each article serves as a

somewhat different use-case involving various distinct data types and circumstances.

**2.3 Work with authors to meet the data-stewardship standards**

A few authors expressed concerns about releasing data provided to them from other scientists or about making digital versions of their output data available. These views were among those revealed by extensive surveys that explore data-sharing practices and attitudes among scientists (Stuart et al., 2018; Schmidt et al., 2016; Tenopir et al., 2011) and are

addressed in the interactive discussion[11] associated with this note. The vast majority of authors were agreeable to making the modifications specified by the editorial team. Once manuscripts were revised to address the data-review comments, they were again evaluated by the editorial team, with the primary focus on the data availability statement and the data citations.

---

[13] https://www.climate-of-the-past.net/about/data_policy.html, https://publications.copernicus.org/services/data_policy.html

Papers were accepted only after all of the input and output data had been properly archived and reviewed for completeness by the editorial team. This stage required coordination between the author, editor, publisher, and data repository, with somewhat different procedures used for each case. Some repositories allowed reviewers to privately access submitted data prior to public release, others could set up a shell to receive the data, and the data were circulated by the author to reviewers and the repository in preparation for publication. Whatever the exact procedure, the intent was for authors to make their datasets available to reviewers and to transfer their data to a public repository prior to final acceptance of the article.

For the special issue, we discouraged the use of supplements as the sole data repository. Unlike articles, supplementary files are archived only on the publisher's in-house servers. Many authors did include data supplements, but also archived their data in a long-term data repository. There are several other advantages to doing this: (1) the dataset can be updated and versioned through data repositories, whereas a supplement is static; (2) the data are generally more discoverable and, with the proper metadata, more reusable when curated through a community-based repository; (3) there is an opportunity to credit to data generators whom might not be an author on the publication.

## 3. Challenges

### 3.1 Determining which data are "essential"

There are practical limits to what can or should be made readily open[14]. Deciding which data are important to make available for reuse was not always obvious to us or to the authors. Not all of the data presented in a publication were necessarily essential for reproducing the study results, and some might not be useful for future researchers. While there may be unforeseen potential uses of data, requiring authors to archive data that have little bearing on the primary outcome of a study can be onerous and pointless. Instead, the utility of each dataset was evaluated in context of the unique contribution of the particular study. Discussions among the editorial team, which included disciplinary specialists, were frequently needed to judge the significance of particular datasets and to help authors prioritize their data management effort.

*Output data.* The most important outcomes of the study were usually clear, and the resulting datasets that comprise those outcomes were easy to identify. They constituted the data that future users might need to compare their data with similar data, or to test the sensitivity of the results to different assumptions, or to incorporate the data into a future data synthesis. Some manuscripts included various renditions of the output data, such as the outcomes of different sensitivity experiments or different data-processing routines, such as levels of smoothing. In these cases, the version that was favoured by the author and the one that most likely would be reused in future research was identified as the top priority for archival. Studies often presented multiple datasets as the primary outcome; in this case, the entire suite of data was submitted to a data repository and a single landing page was used to organize the various datasets as a package under one link.

---

[14] https://doi.org/10.5194/cp-2017-157-SC7

In addition to the data themselves (often time-series or spatial data), essential metadata needed to improve discoverability and facilitate the accurate reuse of the dataset were required to be included. What constitutes essential metadata for paleo data varies for different data types and purposes, and is different for data appearing for the first time in a new publication versus data that have been rescued from previous publications but were never transferred to a public repository. An effort to advance paleo-data standards, especially for those associated with paleoclimate proxies, is underway through the NSF-EarthCube-supported LinkedEarth project[15] (Khider et al., 2017). Policies and standards for the availability of computer code are less developed, but are being addressed by some modelling-focused publications (e.g., Copernicus' journal, *Geoscientific Model Development*[16])

*Input data*. Identifying the essential input data used in a study was often less obvious. If a particular dataset was necessary to reproduce the primary results of the study, it was considered essential and therefore necessary to make available through a public repository. Most of the results from previous studies that were mentioned as supporting information, but not analysed or featured graphically, were deemed as non-essential to the primary outcome of a study and therefore a bibliographic citation sufficed. Most papers included a discussion and figure that compared the outcome of the study with previously published data. The original publications were cited, but their underlying digital data were not always available. In this case, the editorial team considered whether the implications of the data comparison were central to the conclusions of the study, or in the case of review papers and data syntheses, whether the data summaries would be valuable for future researchers. If so, the editorial team asked the authors to make the previously published data available through a public repository.

### 3.2 Using data generated by other researchers (third-party data)

Using data that are already available through a public repository is straightforward; relying on non-archived data, in contrast, hinders transparency, reproducibility and reusability. This is common practice in paleo sciences because only some paleo data are presently available through public repositories. The studies describing the data are typically published and therefore the data are technically public, at least in graphical form. Rather than digitizing the data from the original publication and archiving the second-hand, degraded version, the editorial team asked authors to be stewards of the data assets that they used in their study by working with the original data generators to transfer the data and associated metadata to a public repository, with the goal of enhancing their discoverability and intelligent reuse by downstream scientists. In a few cases, the editorial team assisted with this data transfer.

All authors of the special issue agreed to follow the PAGES data policy[8], which requires data to be made available at the time of publication. This policy assures that each dataset has been vetted by peer review and stands on its own merits before it is accepted as input to a subsequent publication. In some cases, third-party data from the public domain that lacked an associated peer-reviewed publication was considered appropriate to include in a study. These datasets were associated

---

[15] http://linked.earth
[16] https://www.geoscientific-model-development.net/about/code_and_data_policy.html

with sufficient metadata for intelligent reuse and were attributable to the original data generator through a proper data citation. Third-party software tools used for data analysis and display were addressed similarly. In the undesirable situation when third-party data are being published for the first time, or for data that reside only in a graduate thesis, we suggest that an option to hold the data in trust by a repository for a limited embargo period could help protect first-use rights of data generators.

In one example of authors engaging data generators, Shuman et al. (2017a) synthesized data from 93 previously published proxy records of North American hydroclimatology, over half of which were not available through public archives. The digital data had been provided by the original data generators for the purpose of the synthesis. The authors compiled the data along with essential metadata gleaned from the published articles describing the datasets. They input the data and metadata using the Linked PaleoData[17] structure (LiPD; McKay and Emile-Geay, 2016). Once contained in LiPD format, the (meta)data could be automatically translated to the template used by the World Data Service (WDS) for Paleoclimatology (hosted by NOAA). These files were then sent to the original data generators for validation before transferring them to NOAA-WDS Paleoclimatology for long-term archival. Each curated dataset received a unique and persistent identifier, which was cited to credit the original data generators. In addition, as part of the contribution of their synthesis, Shuman et al. (2017b) compiled all 93 proxy records, including the metadata needed for accurate and consistent reuse of the data, into a single package of machine-readable files, including several serializations. These were posted on the landing page[18] along with the primary outcome of the synthesis.

Another approach to engaging data generators is a data publication. These focus on the data compilation itself, including a description of the dataset and the procedures used to assemble it, rather than scientific interpretation of the data. Authorship on such products can be inclusive of all those who contribute and validate the data and metadata. The data product can then be cited to credit data generators, and it can be used as the basis for addressing research questions by the community. The PAGES 2k Consortium (2017a) temperature dataset is a recent example.

### 3.3 Understanding how to use data citations

A major motivation for the data stewardship activity behind the PAGES 2k special issue was to foster the use of data citations as a mechanism for giving explicit credit to the data producers, with greater exposure and citation of their work. Data are fundamental products of research; citing their source, like all other sources of information, is good research practice (CODATA-ICSTI Task Group on Data Citation Standards and Practices, 2013; Data Citation Synthesis Group, 2014; Starr et al., 2015). Data citations facilitate open-data sharing and they assign credit to the data generator, which might be someone other than the author of the associated article. While data citations are gaining traction in the research community, most paleo scientists have not used them and most paleo-oriented journals have not yet established specific instructions to authors for their style and use. For the PAGES 2k special issue, authors were asked to include two citations to credit previously

---

[17] http://lipd.net
[18] https://www.ncdc.noaa.gov/paleo/study/22732

published essential input data: (1) a bibliographic reference where the data are described, and (2) a data citation that points to the version of the dataset as it is stored in a public online server. For datasets that relied on data-processing tools to access, plot or analyse the data from a data repository, authors were asked to include a citation to the code or software. Authors were also asked to provide a data citation for the primary outcome of their study.

A data citation is different than a traditional bibliographic reference, which cites the article that describes the dataset. Instead, a data citation refers to the location of the original data as lodged in a pubic repository (e.g., PAGES 2k Consortium, 2017b). At the core of a data citation is a persistent identifier, which is machine actionable and globally unique (Fenner et al., 2016). The identifier is typically a DOI number, or a URL link issued by the NOAA-WDS Paleoclimatology. Other URL links that reference data or data-access tools on the Internet often brake after a short time (Dellavalle et al., 2003) and should not be used. Data contained in supplementary information attached to an article does not suffice as a data citation because it might require a journal subscription to access, and publishing companies are not always recognized as secure long-term data repositories (e.g., a member of the ICSU World Data System[19]). In addition, journals do not provide the same level of curation and standardization for data and metadata as do community-based repositories. Sometimes, more than one article describes the same dataset, in which case a data citation is needed to resolve the provenance of the data. Sometimes, datasets evolve as additional information is gathered and the original data are upgraded or modified. Unlike journal supplements, the major paleo-data repositories offer a means for versioning of datasets, and the version can be specified as part of the data citation, although this practice is and its conventions are not yet well described.

Authors chose different approaches to citing data sources and the editorial team did not prescribe a specific style. Some authors included the DOI (or URL issued by NOAA-WDS Paleoclimatology) identifiers within a table that listed the datasets used in the study. This structured format enables future users to conveniently access datasets directly from a list rather than having to locate the DOI (or URL) identifier within the references-cited section. On the other hand, including the data citations in the reference list makes it easier for publishers to capture and processes the citations in the same way as other references (Cousijin et al., 2017). In the future, it may be possible for automated data harvesters to recognize and resolve DOI (or URL) identifiers from within the text outside of the reference list. Other authors used the familiar in-text call-out (author, year) and integrated the data citation into the references-cited section. Some authors included both an in-text call-out to the data citation and the DOI (or URL) within a table. Although somewhat redundant, this dual approach has merit. As journals move toward enforcing open-data practices, the specific content and placement of data citations will need be made explicit as part of the information to prospective authors.

The primary results of studies in PAGES 2k special issue were lodged at public paleo-data repositories, and data citations, or just the DOI (or URL) identifiers, that point to these new datasets were incorporated in the data availability statement. The data citations for most studies linked to a landing page at either NOAA-WDS Paleoclimatology[20] or

[19] https://www.icsu-wds.org
[20] https://www.ncdc.noaa.gov/data-access/paleoclimatology-data

PANGAEA[21] where multiple input and output datasets are listed. For some of the data-synthesis and review articles, authors added metadata to the input datasets to enhance the re-usability and formatted the data to improve machine readability. Rather than revising the original dataset, the modified datasets were included as part of the original contribution of the synthesis study and were listed on the landing page for that study, along with the major outcomes of the synthesis,

sometimes in more than one digital format.

## 4. Outlook

Open-data sharing will accelerate as the advantages become more obvious and as practices that reduce the loss of valuable data become widely expected. The advent of metrics that track when and where data have been cited and reused will encourage data sharing by enabling data generators to quantify the impact of their scholarly product. Data stewardship is a

natural element of the publication process when scientists often wrap up a study, and editors and reviewers have an opportunity to put established open-data policies into practice. Additional work is often required to prepare data with sufficient metadata for archival, but the effort is relatively modest compared to the work involved in the publication itself, and in consideration of the long-term benefits of data stewardship. Keeping the focus of data sharing on the essential data that are most likely to be useful for future studies helps to avoid unnecessary effort for the sake of following a blanket open-

data policy. Considering the wide variety of paleo-data types and the recency of open-data policies, more examples like the PAGES 2k special issue will help editors and authors to navigate the ambiguities and challenges associated with implementing data best practices.

**Team members – PAGES 2k special-issue editorial team**

Nerilie Abram, Research School of Earth Sciences, Australian National University, Canberra, Australia.

Michael N. Evans, Department of Geology & ESSIC, University of Maryland, USA.

Pierre Francus, Institut National de la Recherche Scientifique, Centre Eau Terre Environnement, Québec, Canada and GEOTOP Research Center, Montréal, Canada.

Hugues Goosse, Earth and Life Institute/Georges Lemaître Centre for Earth and Climate Research, Université Catholique de

Louvain, Belgium.

Hans W. Linderholm, Regional Climate Group, Department of Earth Sciences, University of Gothenburg, Sweden.

Marie-France Loutre, PAGES International Science Office, Bern, Switzerland.

---

[21] https://www.pangaea.de

Belen Martrat*, Department of Environmental Chemistry, Institute of Environmental Assessment and Water Research, Spanish Council for Scientific Research, Barcelona, Spain.

Helen V. McGregor, School of Earth & Environmental Sciences, University of Wollongong, Australia.

Raphael Neukom, Oeschger Centre for Climate Change Research & Institute of Geography, University of Bern, Switzerland.

Scott St. George*, Department of Geography, Environment and Society University of Minnesota, Minneapolis, USA.

Chris Turney, School of Biological, Earth and Environmental Sciences, University of New South Wales, Sydney, Australia

Lucien von Gunten, PAGES International Science Office, Bern, Switzerland.

* Data reviewers in addition to the official co-editors of the special issue.

*Acknowledgements*. We thank the authors of the PAGES 2k special issue, Copernicus Publications, and the paleo-data repositories who worked with us to develop practices to help reduce the loss of valued data. We are grateful to those who reviewed an earlier version of the paper and contributed to the frank debate in the interactive discussion, including its

spin-offs on social media. This paper is a contribution to the Past Global Changes (PAGES) Data Stewardship Integrative Activity. PAGES is supported by the US and Swiss National Science Foundations, and the Swiss Academy of Sciences. DSK is funded by NSF-AGS-1602105.

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

**Figure 1.** Summary of editor and reviewer roles in implementing open data as part of publication.

| | Reaffirm the goals of relevant policies | Review manuscripts for data accessibility | Work with authors on data stewardship prior to acceptance |
|---|---|---|---|
| **Major steps** | - specify what is required vs what is recommended<br>- provide examples of best practices<br>- require 'data availability' section with submission | - data availability section<br>- tables and figures<br>- **input data:** data needed to validate the primary conclusions<br>- **output data:** primary outcomes, especially those favored by the author | - review revised version of datasets and data citations<br>- confirm transfer of curated data to repository |
| | Understanding how to cite data | Determining which data are essential to cite and archive | Using data generated by other researchers |
| **Challenges** | - distinction compared to bibliographic citation<br>- components, placement | - evaluate in context of study's unique contribution<br>- consult discipline experts | - facilitate data rescue<br>- address ownership, credit and timing issues |