# Peer review of "Technical Note: Open-paleo-data implementation pilot – The PAGES 2k special issue"

_Climate of the Past, 2017_

## Short Comment (SC1) · 29 Dec 2017

This manuscript raises several important issues around data stewardship that will likely resonate with many practitioners within the paleoclimate community. A recent attempt to articulate best practices regarding data curation, synthesis, and dissemination was provided by the paleo-ice sheet/sea level communities (PALSEA; another PAGES working group) in this journal (CP, v. 12, p. 911, 2016). While echoing several of the recommendations made by the PALSEA group (e.g., on giving proper credit to data creators), the present manuscript adds significant aspects such as the implementation of data review as a key element of the peer-review process. With proper guidelines such as

those put forward here, the pace of data release is likely to accelerate. This should help open the door to future discoveries that were previously out of reach.

---

## Short Comment (SC2) · 13 Feb 2018

Comment on "Open-paleo-data implementation pilot - The PAGES 2k special issue"

David J. Karoly1,2, Kathryn J. Allen3 and Patrick J. Baker3

1School of Earth Sciences, University of Melbourne, VIC 3010, Australia
 2ARC Centre of Excellence for Climate System Science, University of Melbourne, VIC 3010, Australia 3School of Ecosystem and Forest Sciences, University of Melbourne, Richmond, VIC 3121, Australia

Introduction

[Figure]

The authors of the Technical Note manuscript have sought to explain the procedure they claim was used to attain a consistent level of data stewardship across a special issue of the journal Climate of the Past. We are the senior authors of one of the papers published in this special issue (Freund et al., 2017) and we disagree with a number of statements in the manuscript, believe that there are important omissions in describing the procedures that were used, and disagree with the interpretation of the journal's Data policy. The early career researchers who are authors on our paper chose not to be authors of this Comment because they are concerned that the feedback provided in this Comment might damage any future interactions that they wish to have with some members of the PAGES 2k research community.

We very much appreciate the need for input data to be made publicly available, but have a number of concerns about the procedure used for this special issue, its inflexibility, apparent inconsistency and inefficiency. While we agree with the authors that open-data sharing in paleo sciences is likely to accelerate, it is important to discuss the disadvantages as well as the potential advantages of open-data sharing. We are taking this opportunity to comment on the procedure used to attain a high level of data stewardship more generally and the challenges around this.

Specific concerns

Page 1, line 16: The first statement of the Introduction states ". . . the practice of making data readily available is rarely embraced". This is an inaccurate assessment of the practice of making scientific data available in the palaeoclimate and wider climate communities. NOAA has very extensive palaeoclimate datasets made publicly available (https://www.ncdc.noaa.gov/data-access/paleoclimatology-data/datasets ). In our study, almost but not all the proxy data we used were from tree-ring analysis and had been publicly through this NOAA database for many years.

Page 1, line 23: ". . . top-down mandates alone are unlikely to foster the necessary cultural changes in scientific communities." We completely agree with this statement,

but a top-down approach was exactly what was used to implement the data stewardship procedures for this special issue.

Page 1, line 24: "Bottom-up motivation from the research community, including early-career scientists, is needed to drive the open-data revolution" Again, we completely agree with this statement. However, there appears to have been little attempt to have bottom-up involvement of potential authors for the special issue in the preparation of the data stewardship procedures, as far as we know. In fact, exactly the opposite occurred, with some early career researchers given no flexibility when the data stewardship procedures were imposed by Darrell Kaufman and the PAGES 2k special-issue editorial team (hereafter referred to as 'the Team').

Page 2, line 27: The statement that a notice was sent to prospective authors of the special issue to explain the process for data stewardship is misleading. It does not mention the time at which this notice was sent relative to the deadline for submission of papers. The Rules for the special issue were distributed to all planned authors in March 2016.

There was no mention of special data stewardship procedures in these Rules and the only information was that all manuscripts must "comply with the same quality standards as regular submissions". The deadline for submission of manuscripts was 31 December 2016. The data notice was sent not only to prospective authors, but also to authors of submitted manuscripts, in March 2017, more than two months after the original deadline for submissions. The content of this notice is reproduced below. Providing information on data stewardship after the deadline for submissions is inappropriate and a clear demonstration of the top-down process imposed for data stewardship in this special issue.

Dear PAGES 2k Special Issue corresponding authors:

We wanted to remind all authors about Climate of the Past's data policy, which you can find here. PAGES is dedicated to best practices in data stewardship, one of its four

integrative activities, and the editors are committed to ensuring that the Special Issue models these practices. More specifically, all of the essential data used in the analyses and generated as results must be made available through a public data repository where they can be tracked using a "Data Citation," described here. A Data Citation is in addition to a conventional reference to the publication where the data are first described.

We realize that some papers have already been submitted that do not fully meet this expectation, and we don't want to discourage on-time submissions on Friday. Details of the data citations and other aspects of data stewardship will be addressed as part of the regular open manuscript review period, with input from the editorial team. If, however, you or your co-authors foresee any issues with complying with the CP data policy, we regretfully suggest that you withdraw the paper.

Please let know if you have any questions at this stage.

Thank you, Darrell on behalf of the PAGES 2k Special Issue Editorial Team

This notice refers to the Climate of the Past Data policy, from which an extract is reproduced below, from https://www.climate-of-the-past.net/about/data_policy.html Climate of the Past Data policy "Copernicus Publications recommends depositing data that correspond to journal articles in reliable (public) data repositories, assigning digital object identifiers, and properly citing data sets as individual contributions." . . . "To foster the proper citation of data, Copernicus Publications requires all authors to provide a statement on the availability of underlying data as the last paragraph of each article (see section data availability)." Statement on the availability of underlying data Authors are required to provide a statement on how their underlying research data can be accessed. . . . If the data are not publicly accessible, a detailed explanation of why this is the case is required.

While the notice from the Team to authors states that all essential data "must be made available through a public data repository", the Climate of the Past (CoP) Data policy

only "recommends depositing data that correspond to journal articles in reliable (public) data repositories". There is a clear inconsistency between the journal's Data policy and what was implemented by the Team for the special issue. No option was provided by the Team to allow for publicly accessible data sets with a doi through journal supplementary material, which is allowed by the journal. No option was provided by the Team to allow for a detailed explanation of why the data are not publicly accessible, which is allowed by the CoP Data policy. Hence, we believe that there were clear and demonstrable inconsistencies between the journal's established Data policy and the data stewardship procedures implemented by the Team for the special issue, despite the claims in this manuscript.

Page 3, line 10: The statement that "Authors were asked to transfer all datasets that were not easily accessible to a community-recognized public data repository" is misleading. Authors were required, not "asked", to transfer all datasets under threat that their manuscript would have to be withdrawn if they did not comply, based on the data notice distributed by the Team. This top-down approach, with the threat of withdrawal of the manuscript, could be interpreted as intimidation by Darrell Kaufman (author of all the emails) of the graduate student leading our paper, given the inconsistencies between the journal's Data policy and the data stewardship procedures imposed.

Page 3, Section 2.3: This section provides another misleading description of the procedures used to impose the data stewardship procedures on our paper in the special issue. Our paper is a multi-proxy synthesis involving primarily publicly available proxy data in public data repositories. In addition, a small number of the proxies used were new and provided by other researchers, or were publicly available but not in data repositories. Two examples of the interactions between Darrell Kaufman and two data providers are contained in an Appendix to this Comment. As can be seen, this was neither a helpful nor a bottom-up process.

Page 7, Section 4 Outlook: The most important lesson to learn from this special issue is that it is vitally important to distribute the planned data stewardship procedures at

the same time as the Call for Papers. It is also vital to make sure that the planned procedures follow the general interpretation and practice of the journal's Data policy, not an effort to implement a stronger data stewardship policy.

General concerns

While we fully support the call for open data sharing and transparency in the palaeo-climate sciences, we do not agree that publication is always the ideal time to make data available. Specifically, we argue that data availability policies must include some flexibility so as to protect the interests and future career prospects of early career scientists and the longer-term viability of the research groups producing the palaeoproxy records.

The impact of rigid data policies formulated in a top-down manner by experienced researchers (often those involved in modelling or multi-proxy synthesis) with large teams will generally be negative on early-career researchers who are often working to schedules around their PhD study and cannot as rapidly produce the final products of their work as can a larger group. With a desire to succeed and contribute to the science, this leaves them vulnerable to 'scientific exploitation' and, in more serious cases, may compromise the successful completion of their postgraduate studies and future careers. It is these people in particular, that data policies should help protect. A data policy should also respect the fact that, in many cases, funding for the generation of new records comes from agencies (Government and NGOs) that expect a return on their investment from the funded group, not from a different group. Rather than a didactic top-down approach from data modellers or synthesizers who clearly require palaeo-proxy data for their use, a data policy should be formulated with input from those who generate the records.

At the time of the Call for Papers for the CoP special issue, it was not apparent that all data would be required to be made publicly available upon publication of papers, nor that the policy for the special issue would differ from that for Climate of the Past

generally. We are aware of at least one author who withdrew their paper due to this not being made clear at the outset. Certainly, if we had been aware of the strict 'new policy' and its heavy-handed application, we would not have included some data sets that were generously shared with us on the basis that they not be made publicly available yet. The group who generated these data sets was working towards their publication as part of an ongoing broader grant-funded project. This situation unnecessarily placed the PhD student writing this paper in an extremely difficult and stressful position.

Reference Freund, M., B. J. Henley, D. J. Karoly, K. J. Allen and P. J. Baker, Multi-century cool and warm season rainfall reconstructions for Australia's major climatic regions. Clim. Past, 13, 1751-1770, 2017. https://doi.org/10.5194/cp-13-1751-2017

Appendix

Example 1

Extract from an email from a data provider in response to Kaufman: "Releasing data from current projects that have yet to finish (or for unfunded projects) runs the real risk of being mined and so diluting my chances of future support. I have been burned before by releasing data. I'm sure this issue has been raised by others.

What particularly gets to me in all of this though, is that when Mandy, an early career scientist, asks for my data and I provide it, I am then the one being portrayed as the villain! I have asked for nothing in return for sharing the data with Mandy-neither co-authorship, nor restrictions on the use of the data. My only request is that a few of the chronologies that we provided be temporarily withheld from lodging with NOAA until after I have been able to publish something myself."

Email from Darrell Kaufman From: Darrell S Kaufman <Darrell.Kaufman@nau.edu> Sent: Saturday, 29 July 2017 10:52 a.m. Subject: PAGES /Freund data request

Dear XXXXX I am writing in regard to a manuscript now under review at Climate of the Past by Mandy Freund, Benjamin Henley, David Karoly, Kathryn Allen and Patrick

Baker, which is part of a special issue featuring the results of the PAGES 2k project. The special issue is also a contribution to the PAGES Data Stewardship Integrative Activity. As such, all of the papers have been reviewed by the 2k special issue data review team in an effort to attain a high and consistent level of data stewardship across the volume. This involves enacting the publisher's data policy of including a 'data availability' section in each paper, which specifies where the essential data used in the study are located. If extenuating circumstances preclude the public release of data used in a paper, the reason must be clearly stated. In addition, all authors of the special issue are using data citations so the source of the data can be tracked and attributed to the data generators.

The rainfall reconstruction in Freund et al.'s paper was based on a large number of proxy records, including several of yours. My understanding is that the data that you provided the authors have not been submitted to a public repository and that you instructed the authors to not release the data as part of their larger synthesis.

The special issue data review team requests that you reconsider your position. Best practices dictate that all data used to formulate the major results of any study be made available along with the publication to assure reproducibility. As a signatory on the International Accord on Open Data, PAGES is committed to a promoting a high level of data stewardship across its activities. Progress within our community is advanced when data are made available for reuse.

The data you provided to Freund et al. have already been formatted by the authors in a way that can generate a NOAA-Paleoclimatology .txt file. We can facilitate the transfer of your data to NOAA or other long-term archive so that you can receive a persistent identifier, which can be used for a proper data citation in your name.

Can we proceed to help you transfer the dataset to a public repository so that it can be properly cited by Freund et al. and so this PAGES Data Stewardship Activity can achieve its goal?

Thank you. Darrell

Darrell Kaufman, PAGES Executive Committee On behalf of the PAGES 2k Special Issue Data Review Team (Belen Martrat, Scott St. George, Nerilie Abram, Raphael Neukom, Marie-France Loutre, Lucien von Gunten)

Example 2

Information from a data provider on Siple Dome shallow ice core B water isotopes: The authors should cite Jones et al., 2014 (https://www.clim-past.net/10/1253/2014/cp-10-1253-2014.pdf). In that manuscript, the isotopic data for Siple Dome shallow ice cores B-H is provided in a supplemental file (http://dx.doi.org/10.5194/cp-10-1253-2014-supplement).

Response from Kaufman: Climate of the Past discourages the use of supplemental files for long-term data archival and the publisher (Copernicus) is not an officially recognized data repository. Best practices would be to transfer the data to NOAA Paleo or similar and to obtain a persistent identifier for a proper data citation.

Please also note the supplement to this comment:
https://www.clim-past-discuss.net/cp-2017-157/cp-2017-157-SC2-supplement.pdf

---

## Short Comment (SC3) · 13 Feb 2018

I commend and fully support David Karoly, Kathy Allen and Patrick Baker for their very detailed and completely justified critique of the Technical Note: Open-paleo-data implementation pilot. It is quite clear that the ramifications of this "pilot" have not been thought out well by its authors, especially given the way that it forces graduate students and early-career scientists to give up their sensitive new data prematurely before their degrees or projects are completed. I know because this is a very real concern of my two graduate students and they deserve to be concerned given this so-called "best practices" data stewardship policy that prompted the earlier comment. The same issues apply to funded research projects that may have very good reasons to not want to release data before their projects are completed. Depending on the funding source, this may not even be allowed. Does this mean that cutting-edge research cannot be published in certain journals before the end of a project when the data will be released? Of course, we all want to see data placed in a public domain data base like the ITRDB, but there has to be a better middle ground way of doing this than that proposed in the Technical Note. One that better respects those who generate the data, and one that is not unreasonably forced upon vulnerable graduate students and early-career scientists as the case is now. I look forward to discussions that would move us towards a more reasonable middle ground of data stewardship policy.

———————————————

---

## Author Comment (AC1) · 21 Feb 2018

**Reply to comment by Karoly and others**

D. Kaufman and the PAGES 2k Special Issue Editorial Team (Nerilie Abram, Michael N. Evans, Pierre Francus, Hugues Goosse, Hans Linderholm, Marie-France Loutre, Belen Martrat, Helen V. McGregor, Raphael Neukom, Scott St. George, Chris Turney, and Lucien von Gunten)

We (the PAGES 2k Special Issue Editorial Team) chose the interactive, public-discussion platform of *Climate of the Past* to trial our data stewardship initiative because we recognize that advancing open-data policies requires community discussion. The comment by David Karoly, Kathryn Allen and Patrick Baker was what we were looking for: a frank debate about translating data policies into practice. We thank the commenters for expressing their views and for exposing the contention that data policies often elicit, and we appreciate their general "*support . . . for open data sharing and transparency in the palaeoclimate sciences.*"

The PAGES 2k special issue was an opportunity, for those who opted to participate, to work toward a higher standard of data stewardship, a goal shared by the journal publisher, who endorsed our enforcement of the data policy. The process was new, and everyone (especially us) learned from the experience, including ways to improve. For this reason, we prepared the Technical Note that describes the procedure we used to pilot a strict and systematic interpretation of the journal's data policy within the framework of the PAGES Data Stewardship Activity[1]. The Technical Note was not meant to reiterate the merits of open-data policies, which are described in articles cited in the Note (e.g., the FAIR Guiding Principles[2]). Nonetheless, we agree with the commenters that, "*it is important to discuss the disadvantages as well as the advantages of open-data sharing.*"

**I. The commenters present three disadvantages of an open-data policy:**

(1) *"The impact of rigid data policies. . . will generally be negative on early-career researchers who are often working to schedule around their PhD study and cannot as rapidly produce the final products of their work. . . This leaves them vulnerable . . . and may compromise . . . their postgraduate studies. . ."*

**Response:** We fully agree that the interests of early career researchers (ECRs) must be protected. Whether open-data policies are generally negative for ECRs is debatable: Open data can (and has, based on the experiences of early and mid-career
* * *
[1]http://pastglobalchanges.org/ini/int-act/data-stewardship

[2]http://www.nature.com/articles/sdata201618

scientists involved in the PAGES Data Stewardship Activity) lead to new collaborations, greater impact and citation of their findings, and demonstrated compliance with current university and funding-body data policies, which in turn aids the scientist's case for future employment and funding support. Following open-data best practices demonstrates that an ECR values scientific transparency and reproducibility and has the knowledge to advance this scientific trend.

We understand that releasing data relinquishes control, but we stand with the affirmations of professional scientific organizations that reproducible science requires that the underlying data (and essential metadata) be released as part of a peer-reviewed study. The commenters might be referring to the additional time needed to prepare a dataset for delivery to a repository, relative to the short timeline of a PhD program. We contend that the additional work is minor relative to the effort involved in publishing and that is most efficient when done as part of preparing a publication. The process becomes easier as streamlined workflows are developed.

(2) "*A data policy should also respect the fact that ... funding ... comes from agencies... that expect a return on their investment from the funded group, not from a different group. Rather than a didactic top-down approach from data modellers or synthesizers who clearly require palaeoproxy data for their use, a data policy should be formulated with input from those who generate the records.*"

**Response:** We agree that data generators should not turn their data over to another group in a way that diminishes the credit received for their contribution. We support – and many of us are among – data generators who are pushing for widespread adoption of data citations[3] as a mechanism for data generators to receive credit on par with a bibliographic citation. A major motivation for the data activity behind the PAGES 2k special issue was to foster the use of data citations. The editorial team worked with authors to help them include data citations in their papers. In the case of the dataset referred
* * *
[3]https://www.force11.org/datacitationprinciples

to by the commenters, we reached out to the data generators and offered to *"facilitate the transfer of your data to NOAA or other long-term archive so that you can receive a persistent identifier, which can be used for a proper data citation in your name."* Most of the data generators agreed because this supports, rather than diminishes, the credit that they accrue. The citation, re-use and development of new applications of publicly available results of funded research is in fact generally encouraged, a quantifiable credit to both the investigator and the agency.

(3) "... *we would not have included some data sets that were generously shared with us on the basis that they not be made publicly available yet. The group who generated these data sets was working towards their publication as part of an ongoing broader grant-funded project. This situation unnecessarily placed the PhD student writing this paper in an extremely difficult and stressful position.*"

**Response:** The commenters' paper was more difficult in regards to data stewardship than any other in the special issue because, unlike most every other paper, some of the previously published data used in their study were not available through public repositories. We regret that this process led to a stressful situation. We believe that it could have been avoided through discussions between the data generators and the authors during the data-gathering phase. In this case, it appears that the reluctance of data generators to make their data openly available should have precluded their inclusion into the dataset used in the commenter's paper. We contend that data syntheses need to be based on publicly available data and that studies that rely on unpublished or nonpublic data rightfully risk skepticism, because results cannot be verified. Ideally, a dataset should be vetted through peer review, and stand on its own merits before it is accepted as input to someone else's publication. We recognize that there are cases when it is appropriate for a study to include data from the public domain that lack an associated peer-reviewed publication, especially for relatively simple types of data, or for data that come with sufficient metadata for intelligent reuse and are attributable to the original data generator through a proper data citation. While offering unpublished

data for use in an another study might seem generous, use restrictions attached to the data can make reliance on unpublished data counterproductive. We advocate the use of data-oriented publications, such as *Scientific Data,* as an alternative means for data generators to obtain credit through subsequent citations of their data.

**II. The commenters disagree with our interpretation of the journal's data policy:**

(1) The journal's data policy "*recommends depositing data . . . in reliable (public) data repositories*" whereas we *"required"* this.

**Response:** In our view, following a policy requires adopting its recommendations, and the purpose of our activity was explicitly to apply a strict and systematic interpretation of the policy, consistent with the COPDESS statement referred to in the journal's data policy[4]. More important than semantics, the journal policy states, "*If the data are not publicly accessible, a detailed explanation of why this is the case is required.*" We conveyed this option to the data generator and to the commenters, "*If extenuating circumstances preclude the public release of data used in a paper, the reason must be clearly stated.*" In response, the data generator, who was not part of the author team, explained, "*Releasing data from current projects that have yet to finish (or for unfunded projects) runs the real risk of being mined and so diluting my chances of future support.*" We did not think that this explanation was appropriate for publication in the data availability section of the article. More importantly, the data in question had already been published and cited as such by the authors, so the issue was the availability of a digital version of data already used in a previous publication, which is difficult to justify withholding. We agree that special circumstances can arise that must be considered on a case-by-case basis and we remained open to such explanations from authors throughout the process.

(2) *"No option was provided by the Team to allow for publicly accessible data sets with a doi through journal supplementary material, which is allowed by the journal."*
* * *
[4]https://www.climate-of-the-past.net/about/data$_{p}olicy.html$

**Response:** Copernicus Publication does not offer data-curation services and supplements are archived only on their in-house servers. Unlike articles, supplements are not sent to long-term archives. The journal's data policy instead asks authors to use a data repository that is registered through re3data.org. When asked about this by the authors during the manuscript review, we clarified the policy.

**III. The commenters take issue with our procedures and assertions:**

(1) The notice of our intention to enforce the journal's data policy was not sent to authors until after some papers had been submitted.

**Response:** We agree that an earlier reminder of the journal's data policy and our data stewardship initiative for this special issue would have been better, and will include this lesson learnt in our revisions of the Technical Note. Of the 17 papers accepted to the special issue, two had been submitted prior to our message conveying explicit instructions to corresponding authors. We assumed that the open-data policies for the special issue were clear from the outset because, beginning with the first PAGES 2k Circular[5] in 2010, and nearly every year since, the PAGES 2k community has recognized the need to make all data used in its products available publicly, a guideline that applies to all PAGES working groups. More importantly, we were unaware that the commenters were not in agreement with our instructions that "*all of the essential data used in the analyses and generated as results must be made available through a public data repository where they can be tracked using a Data Citation.*" In fact, in their reply[6] to our comments, the authors stated that they would "add all data citations" linked to the digital data used in their study. This was reassuring and, in the end, was done by the authors.

(2) The terms "*top-down*" and "*bottom-up*" were misused.

**Response:** We agree that, as editors of the special issue, our role was to require
* * *
[5]http://www.pages-igbp.org/ini/wg/2k-network/intro
[6]https://www.clim-past-discuss.net/cp-2017-28/cp-2017-28-AC3.pdf

authors to follow through on their replies to reviewer's comments, as well as to follow the intent of the journal's and PAGES' data stewardship statements. On the other hand, the PAGES 2k principles of data archiving that we enforced have been developed over many years and in consultation with a large interdisciplinary group of paleo-data generators, and they are shared by the global scientific community. Nonetheless, in the revisions to the Technical Note, we will replace the ambiguous terms with more descriptive text.

(3) "*. . .manuscripts would have to be withdrawn if they did not comply. . .*" with the data policy.

**Response:** We gave authors of the special issue papers the option of complying with the data policy (as applied by the PAGES 2k special issue), transferring the manuscript out of the special issue, or withdrawing their paper. The one paper that was not able to meet the strict open-data requirements being piloted in the special issue was simply transferred out of the special issue and into a regular issue of the journal, and little time was lost.

(4) Our statement that "*the practice of making data readily available is rarely embraced*" neglected to recognize the major efforts to secure our community's data resources.

**Response:** We agree; however, on the basis of the commenters' concerns, it's fair to say that making data readily availability is "not always" embraced. We will revise the statement.

**IV. Key points**

It's important to keep the two issues separate: (1) whether new data should be made available at the time of publication, and (2) whether previously published but not-yet-publicly-archived data should be used in a synthesis study. The datasets referred to by the commenters relate to the second issue. The data had been attributed to previous publications; they were forwarded to the authors for their synthesis study with the

provision that they not be released. For many years, the PAGES 2k community has reaffirmed the requirement that its data products must be based on publicly available data. The authors should not have accepted publicly withheld data into their PAGES 2k data compilation if the data generators were unwilling to publicly archive the dataset.

Regardless of one's views on open-data sharing, reproducibility is fundamental to scientific integrity and progress. It is only possible if the data used to generate a result are publicly available. One goal of the PAGES 2k special issue was to apply a rigorous interpretation of the journal's data policy, an activity agreed to by the participants as evidenced by their response to our comments in the open discussion phase of the paper reviews. Any authors who preferred to not follow the recommended policies could have moved their paper to a regular issue of the journal. Lacking publicly available data, they would not have been considered part of the PAGES product. While we do not endorse looser data polices, we understand that our approach to implementing them is not yet standard practice in paleoscience.

The intent of our implementation pilot and its documentation in the Technical Note is to help move the community in the direction of open data. The intent was not to extract data from our colleagues, but to create a product that gives explicit credit to the data producers, with greater exposure and citation of their work. We believe that this approach will promote transparency and reproducibility and will help accelerate scientific discovery through the advancement of science in emergent transdisciplinary directions. We stand with Copernicus Publications and the many other high-level international signatories in their commitment[7] to data stewardship as articulated by the Coalition on Publishing Data in the Earth and Space Sciences (COPDESS). Our community's "*data are special resources, critical for advancing science and addressing societal challenges.*" We concur with COPDESS that "*scholarly publication is a key high-value entry point in making data available, open, discoverable, and usable.*"
* * *
[7]http://www.copdess.org/statement-of-commitment/

---

## Author Comment (AC2) · 22 Feb 2018

We appreciate the comment from Edward Cook expressing concerns about forcing early career researchers (ECRs) to give up their data prematurely before their projects are completed. Cook advocates for flexibility in the policy that data need to be made public at the time of publication. In contrast, we stand with the affirmations of many professional scientific organizations that reproducible science requires that the underlying data (and essential metadata) be released as part of a peer-reviewed study. We also believe that open-data practices benefit ECRs because they can lead to new collaborations, greater impact and exposure, and because best practices in data stewardship

reflect positively on an ECR.

We are not aware of a situation in the PAGES 2k special issue that required ECRs to give up their sensitive data. The situation described in the Karoly et al. comment relates to data that were attributed to a previous publication, but not publicly archived, then used in the commenters' synthesis study years later. Authors who did not want to follow the recommended data policies could have moved their paper to a regular issue of the journal (for a more complete explanation, see our reply https://doi.org/10.5194/cp-2017-157-AC1).

We believe that scooping of data in paleoclimatology is rare and that its risk is outweighed by the scientific benefits of data sharing. If, however, further discussion reveal that open-data policies truly pose a significant obstacle for ECR's pursuit of scientific careers, then we will be among those to search and advocate for solutions. For example, one option that could be explored as a step toward alleviating concerns about releasing ECR data prematurely is to attach an "ECR-data notice" to datasets submitted to public repositories. The notice would alert potential data users that the data are part of an ongoing ECR project. It could solicit new collaborations with other scientists who are interested in the data and provide contact details for the data generator. While this approach does not guarantee that data will be used by others only within the time frame specified, we suspect that it would help to avoid inadvertent use of data that might conflict with the near-term plans of an ECR. There are issues related to this approach that would need to be considered and we encourage further discussion on this topic.

---

## Author Comment (AC3) · 22 Feb 2018

Darrell Kaufman and PAGES 2k special-issue editorial team

darrell.kaufman@nau.edu

We appreciate the positive reaction to our Technical Note. We support the strategies and perspectives used to create the palaeo-sea-level and palaeo-ice-sheet databases, as articulated in a recent contribution to this journal (http://www.clim-past.net/12/911/2016/). Among other salient points, that paper highlighted the importance of crediting data generators. We see data citations as an important development in this regard, and a major element of the data policy we put into practice with the PAGES 2k Special Issue.

---

## Short Comment (SC4) · 23 Feb 2018

The basic principle underlying this PAGES data stewardship activity is that publications need to be accompanied by publicly available data. Not everyone agrees with this first step, making it difficult to resolve the countless secondary issues, individual cases, and permutations that follow. All journals promote data archiving, but only a few enforce it. No wonder; data policies are sticky for both authors and editors.

Where do we go from here? Are the current data-publishing practices optimal, or have we arrived at a point in the evolution of our science and technology when our community decides that it values its data resources enough to make open-data practices

the norm? Is there agreement that the added long-term, community-wide benefits outweigh the additional burden? There will always be reasonable exceptions, but is it time now for us to fully commit?

I agree with Denis-Didier Rousseau, co-editor in chief of this journal, who has encouraged readers to take advantage of this public platform for an open discussion on this crucial issue.

---

## Editor Comment (EC1) · D.-D. Rousseau (Editor) · 23 Feb 2018

I would like to sincerely thank all the contributors of this particular discussion which is reaching the goal we had in mind when launching this journal some almost 13 years ago. This is exactly how we thought intelligent discussions could improve our field through many ways, and that would benefit our community. This is the kind of leadership we were claiming when creating Climate of the Past. Therefore, one can be contradicting the authors of a paper, as long as the debate remains on a fair landscape, this is perfect and remain positive still.

In the present case, critical issues are raised and I am very pleased by the tone of the

discussion. I am very much looking for reading the reviews which should be posted soon to complement the discussion. This is the reason why I strongly encourage the readers, our community, to take advantage of the extended discussion to post other comments and remarks so that by the end we could get a published paper which could be considered as a cornerstone in this crucial issue that open-paleo-data is.

denis-didier Rousseau

Climate of the Past co-editor in chief

---

## Short Comment (SC5) · 27 Feb 2018

This Technical Note raises a number of issues regarding the Open Data movement and data stewardship in the palaeosciences.

I wish to raise two additional points; i) the appropriate time in the data cycle to consider data management and stewardship, and ii) the use of embargoes to assuage concerns regarding data sharing and archival.

Kaufman et al suggest that "[p]ublication is the ideal stage in... a research project for concerted data management and stewardship." This, they claim, is because

"[s]cientists are most familiar with the specific details of their data when submitting a manuscript for publication and therefore are better able to prepare the data for transfer to a public repository at that moments rather than following publication when the familiarity and incentive have faded." I agree that archiving data post-publication is far from ideal, but I disagree with the statement that the point of submission is the ideal time to address data archival and options for stewardship.

The time point when researchers are most familiar with data is, contrary to Kaufman et al's suggestion, at the point of data capture or collection into a digital format. If the data product is the result of subsequent processing of raw data, then it is at this data processing stage when scientists are most familiar with the data product and how it was produced. At this point, not the some later period closer to publication, it is appropriate to prepare data for subsequent archival via the addition of metadata, description of the data, and preparation of any data-processing code. Whilst archival itself may happen at a later stage, "concerted data management" is best conducted as the data are collected, produced, and prepared for analysis (see, for example, Strasser et al "Primer on Data Management: What you always wanted to know" at https://www.dataone.org/sites/all/documents/DataONE_BP_Primer_020212.pdf).

The most important aspect of data stewardship during the transition to Open Data within the palaeosciences is to actively pursue and promote data archival in standard formatss with appropriate metadata and computer code. Indeed, it is good data management practice, as discussed above, to prepare data for archival at the point of creation not publication. However, there is no reason why data cannot be archived in an approved repository prior to publication and held under embargo until a predetermined time. It is not clear from the Discussion Paper whether such an option was considered by the PAGES 2K special-issue editorial team? Such an embargo would allow for approved reuse within a synthesis activity yet retain some control over availability until other activities are complete. Once the agreed embargo period is complete the data should automatically be made publicly available. In the interim, the data could be referenced via a DOI or similar identifier, but not accessed without permission from the data generator. Journals and their editorial boards should determine, in consultation with the community, suitable time limits and other conditions for the embargo, the situations in which embargoes may be used, and the technical arrangements that insure timely, automatic public access following lifting of an embargo.

This should go a long way to satisfying concerns regarding the impacts on ECRs or research projects of participating in open synthesis activities — there are time lines for completion of degrees and projects that would set reasonable limits on any embargo — as well as those pushing for better data stewardship — the data would be archived to a required standard prior to their use in syntheses, for example.

I believe Pangaea is set-up in such a way that this may already be possible, though I suspect some work may be required at the backend to provide the protection of an automatic lifting of an embargo — although the Data submission page of the Pangaea wiki (https://wiki.pangaea.de/wiki/Data_submission) speaks of "publication" in the described workflow, the process seems sufficiently flexible enough to accommodate a general embargo period. To this end, engaging all members of the community, not just the data generators and those organizing large collaborative data syntheses, would be productive in finding workable solutions to the concerns of colleagues in the short-term.

I am no fan of embargoed scientific outputs in general, however it is vitally important that we bring the community with us, not kicking and screaming, but because they want to make the journey. The concerns among some members of the palaeo community could be addressed through appropriate and limited use of embargoes during this period of transition.

---

## Short Comment (SC6) · 1 Mar 2018

The discussion on open-data is essential not only in paleoclimatology but in climate-science more generally. A number of criticisms, allegations, and attacks against climate scientists based and base on a real, perceived, or simply asserted lack of openness.

Reproducibility and validation of results by others relies on clear communication of methods and, often, on the availability of code and (input and output) data.

Let me start with a disclaimer: I am part of the Pages 2k coordinators team. As such

[Figure]

I was shortly involved in the initial discussions of the data-policy for the Special Issue and involved myself recently in discussions about the comments on the Technical Note among the SI editors, the data policy team, and the Pages 2k coordinators. Furthermore, I am not a data-producer (as defined so far in the discussion) but a data-user. If I became a data-producer, it would likely be of model-simulation-output.

I am one of those persons reluctant to publish their data because of, in short, losing control of the data, which however entails various only vaguely conscious and vaguely formulated concerns.

In the context of the discussion on the Technical Note, I would like to very shortly comment on a number of points.

There is an ideal world where every manuscript is open access, the methods used in the manuscript are openly accessible, i.e., the code for the analyses is available, and the input data of the manuscript as well as the output data are public as long as they may be input for new studies.

This does not describe the current state of scientific publishing and I think it is unlikely that it will become the state of things anytime soon. But some relevant issues originate from this idea.

In this ideal, the idea of publishing the data would define our workflows to some extent. That is, when a data set approaches the state in which it could be published, the workflow would include a step of putting the data into its publishable format, preparing the meta-data, and so on. Thus, if we aim at reproducible research, this data-preparation step has to become an essential part of our workflows, and it's current absence - at least in my workflow - is one of the major hindrances in being truly open.

If this becomes part of the workflow, publishing the data is little more than pushing a few buttons. Then researchers can decide themselves when it is the correct point in

time to publish.

This ideal world assumes it is desirable to make data publicly available and thus to lose control. If this is the case, projects and grant proposals include the publishing of the data as an integral part and are structured in a way that it is done without harming the project or the researchers. This harm-prevention implies that the ideal world includes a mechanism to provide the meta-data of a dataset to the public but to still withhold the data for a certain period of time.

As a side-note, I, personally, would be more afraid of scoops where someone does similar analyses and comes to similar conclusions based on a different data-set than that someone uses my data.

Another point is, which output data we really consider to be essential. Someone once stated, every data that is visualized in a manuscript should be available to reproduce the plots. My personal feeling is this is excessive. Is the output of a principal component analysis on an available input data set essential? Is a transformation of an input data set essential? Is the combined transformation of more than one input data sets essential?

This prompts the related question of whether it is better to provide the code or the output data, or whether it is indeed necessary to provide both.

Thus what is essential for reproducing an analysis and essential for subsequent studies?

On a slightly different note, I am not sure to what extend the discussions on the data-policy also considered the output of climate simulations. While many of these are routinely archived at relevant data centers, there are many other simulations which are not archived in an easily accessible manner. They may even only be stored on a collection of external hard drives. Is it enough to put

the subsets used in a manuscript in a repository or should a larger part of the output be publicly available. More generally, how do we deal with large data sets.

Adding to this, if this was a review of the Technical Note, I would recommend to discuss in (even) more detail the following three topics, (i) the concerns on the procedure of the data team as expressed in the discussion so far, (ii) recommendations on work flows and tools supporting the publication of new or previously unavailable data sets, and (iii) specifics for various different data types, e.g., proxies, proxy collections, model simulations, or ensembles of results.

As a general comment, I think there is little doubt that we have to come to a point where we not only encourage making data public but where we as a community take steps to ensure that data is made public. The community as a whole and each individual researcher will likely benefit from the openness of their colleagues more than from withholding their own data. The additional steps in workflows require additional efforts, but I assume these are not bigger than learning to use a new software or hardware tool.

The discussion so far shows from my point of view that while it is - as Kaufman (2018, https://doi.org/10.5194/cp-2017-157-SC4) puts it - indeed time to commit to making data open, there is work to do. We need to be more specific about the how and what and we have to provide mechanisms of harm-prevention not only but especially for early career researchers.

---

## Referee Comment (RC1) · Anonymous Referee #1 · 8 Mar 2018

This is not technically a scientific paper, but more a technical note/review. However, it is very powerful and brings to light three key issue in today's research world and that is, 1) when and how should data that support a paper be made accessible to the research community, 2) how do you cite data, and 3) how long should that data be archived for.

This can be reviewed in two contexts: Firstly, the general principles of science and secondly the context of climate science.

1) The general principles of science. I am a classically trained scientist and it was instilled in me that a true scientist is one whose data and conclusions can be independently verified. That is, when a paper is going through the review process, the data that

the paper is based on, should be made available for review so that it can be determined if the conclusions are based on sound data.

In the 1970s as computerisation slowly took over data acquisitions systems and technologies became more automated, data volumes began to explode and reached the point where they could no longer be published in research papers, mainly because we were still in the era of typesetting and the printing press. This lead to researchers not making it accessible, mainly because it was extremely difficult and personalised data collections became common, which then became regarded as a researcher's competitive edge.

It today's age, with online databases and data systems, this is no longer the case, and data (both raw and highly processed) as well as derivative data products and models should be made available at the time of review to assess scientific integrity of the research paper. Whether the data should then be made more accessible is to some extent dependent on the community norms. Some medical and social science data cannot be made accessible, but the norm is that most research data, particularly that collected with public money should be.

Although this is Copernicus, I note that a similar discussion page is thriving on the AGU web site around the topic 'put it online'.

2) The context of climate science This ideas in this paper are not unique to climate science - in fact many other disciplines are struggling to come to terms with what is proposed in this paper. There are no community norms, although there is evidence that some disciplines are now working together with the publishers to develop best practice guidelines and norms - e.g., in the Earth and environmental sciences ( https://eos.org/agu-news/enabling-fair-data-across-the-earth-and-space-sciences )

As a scientist who has been struggling with this issue for decades, I think that this paper gives an excellent summary of the key ideas and relevant problems. It will become a reference paper for the those that are actually working on trying to solve this issue as it

clearly and succinctly documents the key issues. I can image that it will not be popular as is evidenced by what is appearing on the discussion forum.

Some of the issues the paper is raising are beginning to be addressed in some areas, but these are mostly unpublished and hence cannot be referenced. The advantage of this paper is that it concisely ties all the key issues into a coherent succinct paper that will raise greater awareness of the problems and be of enormous value to those that are now attempting to solve them.

I only found one error that I feel needs to be corrected and that is on line 13 on page 6 where the word 'providence' is used - I think this should be 'provenance'. It is the most minor of errors.

---

## Referee Comment (RC2) · K. Lehnert (Referee) · 13 Mar 2018

This Technical Note is very relevant to advancing open data sharing practices in compliance with the FAIR principles for making data Findable, Accessible, Interoperable, and Reusable. Several initiatives and projects that I am involved with that are pushing the implementation of the FAIR principles for research data forward emphasize the urgent need for scientific communities to come together to develop, articulate, and implement data best practices that are needed for their specific data types to be re-usable and interoperable, and that help make them more Findable and Accessible through the use of recommended metadata and terminology. The author gives a very informative

account of the procedure developed for paleo data and the challenges encountered during implementation for the special issue of this journal. This Technical Note will help other communities to follow a similar approach and hopefully broaden the range of available community-specific standards for data sharing.

My only comment pertains to Page 1, line 26: The author states that "Publication is the ideal stage of a project for concerted data management". Data management and stewardship should be planned at the start of a project and carefully followed throughout the course of the entire project. Agreements on data formats, data provenance documentation, sample identifiers, etc. should be made at the start of a project to ensure consistency of data and metadata, comprehensiveness of metadata, and efficiency of data management. Publication is a stage when compliance with leading practices or standards can be enforced, but it may be too late of the practices were not implemented early on.

---

## Short Comment (SC7) · 30 Mar 2018

I have read this article and discussion with great interest: it speaks to fundamental questions about how we conduct open, transparent, and global-scale earth system paleoscience, in synthesis projects that draw upon the collected data, knowledge, and labor of many (dozens to thousands) of individual scientists, distributed around the world. My perspective is that of a scientist who has worked on many continental- to global-scale data syntheses over the years, as someone not involved in the PAGES 2K initiative, and as one of the leaders of the Neotoma Paleoecology Database (www.neotomadb.org), which seeks to support open data, building communities of Data Stewards, and global-scale paleoscience. Here I first reflect on four emergent themes in this discussion, then briefly note Neotoma's role within the open-data ecosystems that are emerging and the solutions that we are seeking to build.

1) Where are the ethical limits to open data? Open data is clearly a good: it enables transparency, reproducibility, and accountability in science (see comments by Simpson, Bothe, response by Kaufman et al.). Open data enables our field to move from our local-scale records, collected at great effort and cost, to a global-scale understanding of the earth system and its past dynamics. Open data and open workflows accelerate the pace of science and knowledge transmission, by enabling new advances developed by one research team to be quickly adopted by other researchers.

However, there are other goods. Karoly and Cook in their comments raise the important good of protecting early career researchers and their intellectual output. One can imagine an extreme open-data ethos that did not fairly account for this competing good: e.g. requiring all measurements made by an early career researcher to be instantly made available on-line and usable by all. This hypothetical extreme, in which an ECR provided the labor and others instantly reaped the fruits, clearly would carry the open-data ethos too far.

So, we need balancing mechanisms that both encourage open data and create first-use protections for the intellectual work by data generators and early career researchers. Embargoes, as suggested by Simpson, are one important mechanism, and a critical priority for our field should be developing better systems for creating and managing data embargoes. (We are beginning embargo development in Neotoma, see below.) A second mechanism should be to establish different open data norms for primary data papers versus large data syntheses. Primary data papers, that present new data collected and generated by a research team, should be held to a high standard of data openness and publication, with the general expectation that all presented data be contributed to a community open data repository. Large data syntheses, such as the PAGES 2K synthesis, that draw upon both published and unpublished records, need

more of a tiered system that balances open data missions with protections for data generators. Such a tiered system would establish full openness for workflows that use published data, and partial openness for workflows that use unpublished data. Or, efforts such as PAGES 2K may simply opt for simplicity, use published data only, and thereby achieve full openness.

2) Where are the practical limits to open data? This point, raised by Bothe, falls under the general topic of data reduction. In practice we must always curate data and knowledge, in which we make decisions about which data are important and which are not. Some limits relate to data volume, e.g. the question by Bothe about earth system models and how best to store and share their large-volume outputs. Similarly, Bayesian modeling approaches generate large volumes of Monte Carlo Marko Chain (MCMC) traces that are used to generate posterior probability estimates (Blaauw and Christen, 2011; Dawson et al., 2016; Parnell et al., 2016). Should the full MCMC traces be stored, or simply the summary statistics? Some data reductions or transformations are motivated by scientific convention. For example, radiocarbon dates are usually reported as estimated ages, rather than the primary measurements of count statistics for individual isotopes. Some data reductions occur because science data collection operates at the real/virtual interface, with some information easily captured and shared with others (e.g. primary data tables, instrumental outputs, photos) and others less so (e.g. field notebooks, lab notebooks, personal experience, judgment, and decision-making). In general, the advances in data science will make it possible to extend data openness to information that was previously impractical to share (e.g. raw data output from geochemical instrumental systems). But there will always be limits to what can be made readily open, and hence some need for expert judgment and community norms about data curation and sharing.

3) Paleodata are high-effort and therefore high-value. Our proxy records are collected at great cost: they usually require days to weeks of fieldwork in remote locations, months to years of labwork, and months to years of analysis and interpretation by

highly trained experts. Most of this work is supported by public taxpayers via scientific foundations. Often, these data cannot be collected again, for economic reasons (costs associated with field and labwork) and physical reasons (many archives are now lost or in danger of being lost). To me, one of the strongest arguments for open data is as our field's bulwark against data entropy (Michener et al., 1997) and knowledge loss (Jackson, 2012). Our sum total of scientific knowledge can be viewed as a dynamic balance between rates of knowledge and data generation versus rates of knowledge and data loss. The peer-reviewed literature is a long-established and good (but imperfect) system for transmitting and saving scientific knowledge. A grand challenge for our generation is to build equally strong open data systems for transmitting and saving the scientific data that support our knowledge.

4) Good data management begins at point of capture. I fully support Simpson's point that good data management begins at the point of data collection, not at the point of publication. Most of our data losses occur because data management effort is mostly invested at the end of a project cycle, when it is particularly laborious and when scientists are ready to move on to their next paper, grant, or project. Our field needs sustained research and investment in data systems that support and facilitate data management at all stages of the process, with as little burden as possible placed on individual scientists. For example, when coring and drilling lakes, we need integrated data management systems for easily capturing coring metadata, the data and metadata when splitting and imaging cores, the depth models and age-depth models generated from these cores, the proxy measurements made by multiple research groups on these cores, and the eventual analyses and papers that result from these cores.

At Neotoma, our mission is to support global-change research by providing a high-quality community-curated data resource for paleoecological and paleoenvironmental data (www.neotomadb.org) (Williams et al., 2018). We traditionally specialize in paleoecological proxy data from a variety of terrestrial archives (e.g. diatoms, ostracodes, pollen, testate amoebae, vertebrates) and are expanding our data models to store

geochemical data such as stable isotopes and organic biomarkers. We view ourselves as one node in a larger open-data ecosystem, complementary to other primary data archives (e.g. NOAA/NCEI Paleoclimatology, Pangaea) and supportive of high-level synthesis efforts such as PAGES2K, PalEON, or SKOPE. A key element of Neotoma's approach is to stay closely engaged with data generators and data users and to build a network of expert Data Stewards, serving a role akin to editors in peer-reviewed journals. Our governance model is open and built around the concept of Constituent Databases, each representing a particular proxy type or region with associated communities of Data Stewards. Neotoma seeks to enable living data systems, with tools for data updates and amendments by Data Stewards. The most common amendment is addition of new age-depth models, as age-depth modeling approaches improve and data synthesis efforts rebuild age models. But, more generally, we seek to support ongoing improvements to Neotoma's data, mediated by trained Data Stewards, because many forms of data error and corruption are only uncovered by data use.

Neotoma's data use policy includes an embargo policy (https://www.neotomadb.org/data/category/use). For now, policy is ahead of technical implementation: data embargoes are currently implemented by receiving data submissions and preparing them for upload (using the Tilia software system for data cleaning and validation), but avoiding actual upload to the online Neotoma database until embargo is released. We are working on technical implementation of an embargo system for the main database so that data can be submitted to the database and DOIs assigned, but no actual data are exposed or released until the embargo is lifted. The larger goal here is to create systems that encourage good data management practices by data generators (encouraging early data submissions and incorporation of Neotoma into lab-scale workflows) while also protecting the first-use rights of data generators.

A larger and final point is that paleoclimatology and paleoecology have an excellent tradition of data synthesis and data sharing, thanks to our recognition that a globalscale understanding of the climate system demands a pooling of our many site-level records and thanks to pioneering efforts such as CLIMAP and COHMAP. We have a good culture of data sharing, and an awareness of its complexities and tradeoffs, as this discussion by Kaufman and others shows well. Our field is well positioned to create new institutional and social solutions to these new opportunities and challenges of open data sharing, and to be an example to other disciplines wrestling with similar challenges.

References Blaauw, M., Christen, J.A., 2011. Flexible paleoclimate age-depth models using an autoregressive gamma process. Bayesian Analysis 6, 1-18.

Dawson, A., Paciorek, C.J., McLachlan, J.S., Goring, S., Williams, J.W., Jackson, S.T., 2016. Quantifying pollen-vegetation relationships to reconstruct forests using 19th-century forest composition and pollen data. Quaternary Science Reviews 137, 156-175.

Jackson, S.T., 2012. Representation of flora and vegetation in Quaternary fossil assemblages: known and unknown knowns and unknowns. Quaternary Science Reviews 49, 1-15.

Michener, W.K., Brunt, J.W., Helly, J.J., Kirchner, T.B., Stafford, S.G., 1997. Non-geospatial metadata for the ecological sciences. Ecol. Appl. 7, 330-342.

Parnell, A.C., Haslett, J., Sweeney, J., Doan, T.K., Allen, J.R.M., Huntley, B., 2016. Joint palaeoclimate reconstruction from pollen data via forward models and climate histories. Quaternary Science Reviews 151, 111-126.

Williams, J.W., Grimm, E.G., Blois, J., Charles, D.F., Davis, E., Goring, S.J., Graham, R., Smith, A.J., Anderson, M., Arroyo-Cabrales, J., Ashworth, A.C., Betancourt, J.L., Bills, B.W., Booth, R.K., Buckland, P., Curry, B., Giesecke, T., Hausmann, S., Jackson, S.T., Latorre, C., Nichols, J., Purdum, T., Roth, R.E., Stryker, M., Takahara, H., 2018. The Neotoma Paleoecology Database: A multi-proxy, international community-curated

data resource. Quaternary Research 89, 156-177.

---

## Author Comment (AC4) · 7 Apr 2018

We are thankful for the reviewer's insights on the changing nature of data availability. We agree that true science is based on "data and conclusions [that] can be independently verified" and that, in science generally, "the norm is that most research data...should be [made accessible]." We are glad that the reviewer thinks that our Technical Note "will become a reference paper for those...actually working on trying to solve this issue as it clearly and succinctly documents the key issues." And we will correct the typo pointed out by the reviewer.

---

## Author Comment (AC5) · 7 Apr 2018

We thank the reviewer for putting our data stewardship activity in context of the widespread push to implement open data practices and for recognizing the value of our community based effort to trial such practices as part of the publication process. In our revisions, we will place our implementation project into this broader context by pointing out some of the major initiatives that now underway and expanding across the earth sciences to promote open data principles. Prominent among these is the "Enabling FAIR Data[1]" project, an international effort that is moving forward earnestly,

[1]http://www.copdess.org/home/enabling-fair-data-project

with many Earth science journals pledging to shift from open data policies that are 'recommended' to those that are 'required.'

We agree that our statement, "publication is the ideal stage of a project for concerted data management" is only partly correct; sound data management starts with the design of the study and continues throughout the lifecycle of a research project. This point was also expressed in the comments by Bothe, Simpson and Williams. We will rewrite this statement to put the publication step into a larger context. We maintain that publication is a critical, high-value stage for data stewardship and we will strengthen our assertion by adding the following points to the manuscript:

(1) As stated by the reviewer, "Publication is a stage when compliance with leading practices or standards can be enforced. . ." While a comprehensive solution includes incentives and support, as well as enforcement, only funders and publishers have real power to require open data policies.

(2) Although preparing data for archival as soon as they are generated is ideal in many situations, publication is the final pragmatic point in a study to transfer the data to a repository. Familiarity with the data and the incentive to archive them often fade following publication as researchers move on to new projects.

(3) Authors striving to enhance the impact and visibility of their publications are receptive to input from peer reviewers and editors who can help guide authors toward making their data more easily discoverable and reusable.

(4) For many studies, especially in paleoclimatology, the value of the underlying data is strongly related to their interpretation. The most important data and metadata are typically those that are associated with a publication that describes them. Peer review also can aid the interpretation and can help authors to identify the essential metadata. Encoding peer-reviewed expert knowledge into an archived dataset is not possible prior to publication, but is necessary to facilitate the intelligent reuse of the data.

(5) Most paleoclimate studies compare their results with those from related study areas or data types. Most often in paleosciences, the digital data from the previously published studies are not available through public repositories. If the comparison with previous studies is the basis for a major conclusion, such as for a synthesis study, the authors of the succeeding publication can serve as data stewards by facilitating the transfer of data from previous publications to a public repository, with credit given to the original data generator. As part of this data rescue effort, authors can attach relevant metadata to valued previously published datasets to enable their discoverability and intelligent reuse.

---

## Author Comment (AC6) · 7 Apr 2018

The commenter raises two important points: "(i) the appropriate time in the data cycle to consider data management and stewardship," which we address in our reply to referee #2; and "(ii) the use of embargoes to assuage concerns regarding data sharing and archival," which was also discussed in the comments by Williams. We agree that "the concerns among some members of the palaeo-community could be addressed through appropriate and limited use of embargoes during this period of transition." Embargoes were not part of our implementation pilot because authors agreed to follow the PAGES 2k data policy, which requires data to be made public at the time of publication.

[Figure]

Nonetheless, we will include this suggestion in the revised version of the Technical Note, especially as it relates to data that have not yet been published, including those that reside in graduate theses.

---

## Author Comment (AC7) · 7 Apr 2018

The commenter considers a variety of topics, including:

- Which data are essential and therefore important to archive. As discussed in the Technical Note, this was one of the main motivations for, and struggles during, our implementation pilot. Each paper presented unique issues and we hope that the use-case represented by each of the articles in the special issue (available through the CP platform) will help guide future authors and editors.

- How to deal with large datasets, such as climate model output. We did not face

this problem in the special issue. All of the underlying very large datasets used in the papers were already available through existing online servers. The issue will need to be addressed, moving forward.

- Recommendations for data workflows. The publication cited in Simpson's comment, "Primer on data management: What you always wanted to know[1]" includes recommendations and workflows. We will cite this document in our revised paper.
* * ** * *
[1]https://www.dataone.org/data-management-planning

---

## Author Comment (AC8) · 7 Apr 2018

The commenter aptly summarizes the major topics that have been expressed in this interactive discussion, with deep insight from his experience as one of the leaders of the Neotoma Paleoecology Database. We are grateful for his big-picture and constructive comments, and we reiterate his conclusion: "our field is well positioned to create new . . . solutions to these new opportunities and challenges of open data sharing, and to be an example to other disciplines wrestling with similar challenges."

(1) Ethical limits to open data: We agree that the paleoscience community needs to further develop policies and procedures for open data sharing. This includes data em-

bargoes, especially as they relate to data that are archived at the time of capture, prior to publication, and including data in graduate theses. We agree that different standards apply to data that are published for the first time versus those that are rescued from previous publications but were never transferred to a repository, which includes the majority of data in paleosciences. For clarification, PAGES working groups are encouraged to restrict their synthesis products to only data that have already been published, thereby focusing on datasets with peer-reviewed interpretations and honoring the first-use rights of data generators.

(2) Practical limits to open data: We agree that there will always be limits to what can or should be made readily open. From our experience with this open data implementation pilot, deciding which data are necessary to be archived is not always obvious. Not all of the data presented in a manuscript are necessarily essential for reproducing the study results, and some might not be useful for future researchers. Requiring authors to archive data that have little bearing on the primary outcome of a study can be onerous and pointless. Instead, the utility of each dataset must be evaluated in context of the unique contribution of the particular study. There is an urgent need to develop channels for journals to connect directly to disciplinary specialists to guide and model best practices.

(3) High-value data: We agree that developing open data systems that not only preserve the data but also support knowledge generation is a major goal and challenge. Encoding peer-reviewed expert knowledge into an archived dataset is necessary to facilitate the intelligent reuse of the data, but is rarely practiced in paleosciences.

(4) Archiving data at the time of capture: We agree that archiving data as soon as they are generated is often the ideal approach and that support is needed to develop and sustain community data systems that enable this practice. We maintain, however, that publication is the final pragmatic point in a study to transfer the data to a repository. Publication is a critical, high-value stage for data stewardship for the five reasons that we explain in our reply to referee #2.

---

## Author Comment (AC9) · 9 Apr 2018

**Reducing data loss in paleo-environmental sciences**

**by, Darrell Kaufman**

Most articles published today that report paleo-environmental data do not include ready access to the underlying data and essential metadata. This doesn't need to be the norm. The infrastructure is in place to store, access, attribute credit and reanalyze data, and open data policies are already established by funders, journals, repositories and professional scientific organizations. Accelerating the transition to open data

sharing will require more support, training and models of sound data stewardship. In addition, we need community endorsed guidelines to: (1) determine which data and metadata are essential to archive; (2) know which repositories, standards and format to use; (3) apply best practices when using third-party data and software; and (4) protect first-use rights. As journal data policies switch from 'recommended' to 'required', editors and authors need consistent guidelines and channels to connect directly to organized communities of international specialists who can provide responsive guidance and quality control at the valuable publication stage. It will take a major effort and, most importantly, a commitment to shift standard practices to those that reduce data casualties. The practices won't be perfect at the outset, but they are necessary steps. By working together, we can harness more power from our data resources, and we can do it sooner.

The lessons learnt from the PAGES2k open data implementation project, including this interactive discussion, are presently being applied to the PAGES Young Scientist Meeting special issue of this journal. The compendium features articles by early career researchers (ECRs) and is co-edited by a team of ECRs in consultation with senior co-editors. Like the PAGES2k special issue, each article presents a different use-case for determining how to implement data policies. For readers interested in the types and level of guidance provided to authors, the latest data review comment is here, and all data review comments are available through CP's interactive discussions for both special issues.

From my experience with the PAGES YSM special issue, ECRs are eager to practice open data principles. Rather than speaking for them, however, I have asked the new PAGES Early-Career Network for suggestions to improve PAGES' open data policy in ways that would promote career development in the short term, and maximize the scientific benefits of data in the long term. I look forward to their response.

With the close of the interactive discussion, I am grateful to the PAGES2k special issue

editorial team[1] for their valued input, including the extensive deliberations over how to implement open data principles, and over our replies to the comments on this technical note. I am grateful to those who contributed to this interactive discussion, including its spin-offs on blogs and twitter. And, I thank the many journal editors, data repositories, and program managers who have encouraged this activity for their leadership in safeguarding and enhancing our data assets.

[1]Nerilie Abram, Michael N. Evans, Pierre Francus, Hugues Goosse, Hans Linderholm, Marie-France Loutre, Belen Martrat*, Helen V. McGregor, Raphael Neukom, Scott St. George*, Chris Turney, and Lucien von Gunten (*data reviewers in addition to the official co-editors of the special issue)

---

## Author Response (AR1)

**Revisions**

This Technical note generated considerable interactive discussion. Our detailed reply to all the referee reviews and public comments are available as part of the Interactive Discussion. The specific revisions to the manuscript are highlighted as tracked changes in the following pages.

[revised manuscript text omitted]